# Evidence of long-term allocentric spatial memory in the Terrestrial Hermit Crab *Coenobita compressus*

**Ilse Lorena Vargas-Vargas[1], Estefany Pérez-Hernández[1], Daniel González[2], Marcos Francisco Rosetti [2,3], Jorge Contreras-Galindo[4], Gabriel Roldán-Roldán [1] ***

**1** Departamento de Fisiología, Facultad de Medicina, Universidad Nacional Autónoma de México, Mexico City, Mexico, **2** Instituto de Investigaciones Biomédicas, Universidad Nacional Autónoma de México, Mexico City, Mexico, **3** Instituto National de Psiquiatría, Ramón de la Fuente Muñiz, Mexico City, Mexico, **4** ENES unidad Morelia, Universidad Nacional Autónoma de México, Mexico City, Mexico

\* gabargico@gmail.com

## Abstract

Spatial learning is a complex cognitive skill and ecologically important trait scarcely studied in crustaceans. We investigated the ability of the Pacific (Ecuadorian) hermit crab *Coenobita compressus*, to learn an allocentric spatial task using a palatable novel food as reward. Crabs were trained to locate the reward in a single session of eleven consecutive trials and tested subsequently, for short- (5 min) and long-term memory 1, 3 and 7 days later. Our results indicate that crabs were able to learn the location of the reward as they showed a reduction in the time required to find the food whenever it was present, suggesting a visuo-spatial and olfactory cue-guided task resolution. Moreover, crabs also remember the location of the reward up to 7 days after training using spatial cues only (without the food), as evidenced by the longer investigation time they spent in the learned food location than in any other part of the experimental arena, suggesting a visuo-spatial memory formation. This study represents the first description of allocentric spatial long-term memory in a terrestrial hermit crab.

**Data Availability Statement:** All the data included in this manuscript is available at the following links:

## Introduction

Animal cognitive abilities have been shaped by natural selection from the ecological and social challenges each species must contend with, resulting in specialized forms of learning and memory [1]. Although non-associative (habituation and sensitization) and associative (classical and operant conditioning) types of learning are well documented in invertebrates [2, 3] complex forms of cognition such as spatial and episodic memory have been poorly studied beyond cephalopods [4–7] and insects [8, 9].

Spatial learning is a complex ability essential for self-location in the environment and the location of resources in defined time and space. There are two main reference frames of spatial learning: egocentric (sequential) and allocentric (place). The egocentric reference system depends on memorizing the sequence of body movements such as the number of steps and

https://doi.org/10.6084/m9.figshare.22266400
https://doi.org/10.6084/m9.figshare.22259743

**Funding:** This research received financial support from the Programa de Apoyo a Proyectos de Investigación e Innovación Tecnológica (PAPIIT) of the Dirección General Asuntos del Personal, UNAM through grant IN224019 awarded to G.R.R. The funders had no role in study design, data collection and analysis, decision to publish, or preparation of the manuscript.

**Competing interests:** Authors have NO competing interests.

turns as well as their direction to reach a goal, while the allocentric strategy involves remembering the location of spatial cues external to the subject, such as visual and olfactory landmarks in the environment. Allocentric spatial learning has been demonstrated in the octopus [10], cuttlefish [4] and insects like wood ants [11, 12], honeybees [13, 14] and fruit flies [15, 16]. Studies in crustaceans suggest that spatial learning abilities also exist in this group. For instance, evidence of path integration and egocentric spatial learning has been observed in fiddler crabs [17–19] as well as the use of both egocentric and allocentric strategies for maze solving in the crayfish *Orconectes rusticus* [20, 21]. However, examples are still lacking, as to the best of our knowledge, there are no formal studies focused on the evaluation of allocentric learning in Anomuran crustaceans, although previous studies have suggested it might exist [22, 23].

In this study we evaluated the spatial learning abilities of *Coenobita compressus* in a single multi-trial training session, which allowed us to build up a learning curve based exclusively on its working memory capabilities. The task consisted of locating a highly palatable food reward hidden in an open field with a three-dimensional landmark configuration resembling the natural environment of these animals, using visual and olfactory cues for its resolution. Subsequently, we evaluated short- and long-term memory retention without the food reward up to 7 days.

The aim of this work was to determine if terrestrial hermit crabs can learn to solve a spatial task using an allocentric strategy and to determine if they retain this information in the long-term. To do so, we measured (a) the time it took the animals to find (and start consuming) the food reward and (b) the time spent in each of the four quadrants, in which the testing arena was divided, searching for the food reward during short- and long-term memory retrieval tests. We predicted that hermit crabs would locate food faster after repeated training trials but would take increasingly longer time to reach the food location after several unreinforced (extinction) trials. On the other hand, we predicted a longer searching time in the target quadrant during the memory retention tests.

## Methods

### 2.1 Animals

A total of twenty-six pacific sex balanced hermit crabs *C. compressus* (H *Milne* Edwards, 1836; Anomura, Coenobitidae) were hand-collected from sites at Troncones, Guerrero, México (17˚ 47'16"N; 101˚44'17"W). Permission for collection was obtained beforehand from the Secretaría de Medio Ambiente y Recursos Naturales, Subsecretaría de Gestión para la Protección Ambiental (SGPA/DGVS/03187/20). Animals were housed at collective 40-L holding tanks lined with sand from their collection site in low density (10 crabs per tank) for a minimum of 4 weeks prior to testing. Animals acclimated to this system under an illumination cycle of 12:12 (lights on at 7:00 h), a 28˚C temperature and were fed *ad libitum* with hermit crab pellets (TetraFauna), apple slices, drinking water and 1% salt solution. All animals were healthy with intact appendages. The experiments employed a repeated measures design, which required the identities of the test animals to be tracked throughout the experiment. The crabs were individually identified and marked at least one day prior to experiments, with nail polish (enamel) on the shell. No ovigerous females, premolt or recently molted crabs were used. No crabs molted or died during the study. According to Klappenbach et al. [24] satiated animals present a poor food-driven appetitive behavior. For this reason, crabs were unfed 3 days prior the acquisition session to standardize hunger levels across individuals and encourage them to search for food during the training trials. Although animals were housed in groups, they were tested individually in all experiments.

## 2.2 Behavioral procedures

Experiments were conducted in a 30 cm square arena, divided into four quadrants of 15 by 15 cm: upper left (UL), upper right (UR), bottom left (BL) and bottom right (BR), filled with 5 cm of hydrated sea sand. A set up of four objects of different color and size was placed inside. Each object was placed 7 cm from one wall of the arena and centered with respect the two lateral walls. Objects were considered as contextual visual cues. To induce the animals into searching a square plastic tray of 2 by 2 cm with 2 grams of peanut butter was placed in the upper left (UL) quadrant (Fig 1). The amount of food provided (2 grams) was higher than the usual animals' consumption overnight. Peanut butter (Hartleys brand, purchased from a local vendor and stored at room temperature) was used as a reward since it was the most preferred food for crabs in a pilot study we previously conducted on an independent group of experimental subjects. After the tray with the food reward was positioned inside the arena, we placed an individual crab at the starting point. Upon introduction, we restricted the crab's movement by placing a plastic shelter over it for a 1-min acclimation period, so as to minimize inter-individual variation on the initial exploratory activity. After acclimation, the shelter was removed, and the crab was allowed to freely explore the arena for a maximum of 180 sec. Since hermit crabs often respond to overhead movements by retracting into their shells, we were careful not to cast a shadow over them. Observers were not present in the testing room during the trials, and the arena was monitored remotely. The crab's behavior was recorded using a Canon 80D high-definition camera mounted on a tripod without additional lighting. Additionally, the time taken to locate the food was noted down. Trials were stopped when the crab located the food and began to feed, or after 180 sec had elapsed. If a crab located the food, it was allowed to feed on the peanut butter for 5 sec, then gently removed and transferred to a neutral keeping cage covered with a wet sand layer to keep them humid (without food). If a crab failed to locate the food, it was withdrawn and taken to the keeping cage and a time of 180 seconds was noted down. We fixed a 5-minute inter-trial interval during which the sand was rehydrated and mixed-up in order to keep it wet and eliminate possible odoriferous cues left by the subject in the previous trial. All the sessions were video recorded allowing us to analyze the videos offline using Solomon Coder software [25]. Results are expressed as the time to reach the exact place where the food was located, and time spent in each quadrant of the arena (see Fig 1) during the short- and long-term memory retrieval tests.

## 2.3 Training and testing

In this study, we decided to use the standard experimental paradigm for the analysis of allocentric spatial memory in rodents, which fundamentally consists of repeatedly training each experimental subject until reaching a learning criterion, which in our case was the asymptotic decrease in latency to find the reward. Additionally, we measured the distance traveled to reach it. Subsequently, both short- and long-term memory were analyzed by removing the reinforcer and quantifying the latency and distance required to reach the exact location of the reward, as well as the time the subject spent exploring each of the four quadrants of the testing arena. We conducted a total of eleven consecutive training trials on Day 1. In the first three trials the starting point was in the BR quadrant and the food was located opposite to it, in the UL. In order to facilitate the location of the food, the tray with the peanut butter was placed at the level of the sand in such a way that it was visible when the crab had overcome the landmarks in the arena and was inside the target quadrant (UL). In these three trials animals were examined for behavioral reactions to the novel food reinforcement (peanut butter). Crabs were observed during feeding to ensure they were showing foraging behavior [26] and consumed the food. In the next eight trials we evaluated whether animals' performance improved over time. In these,

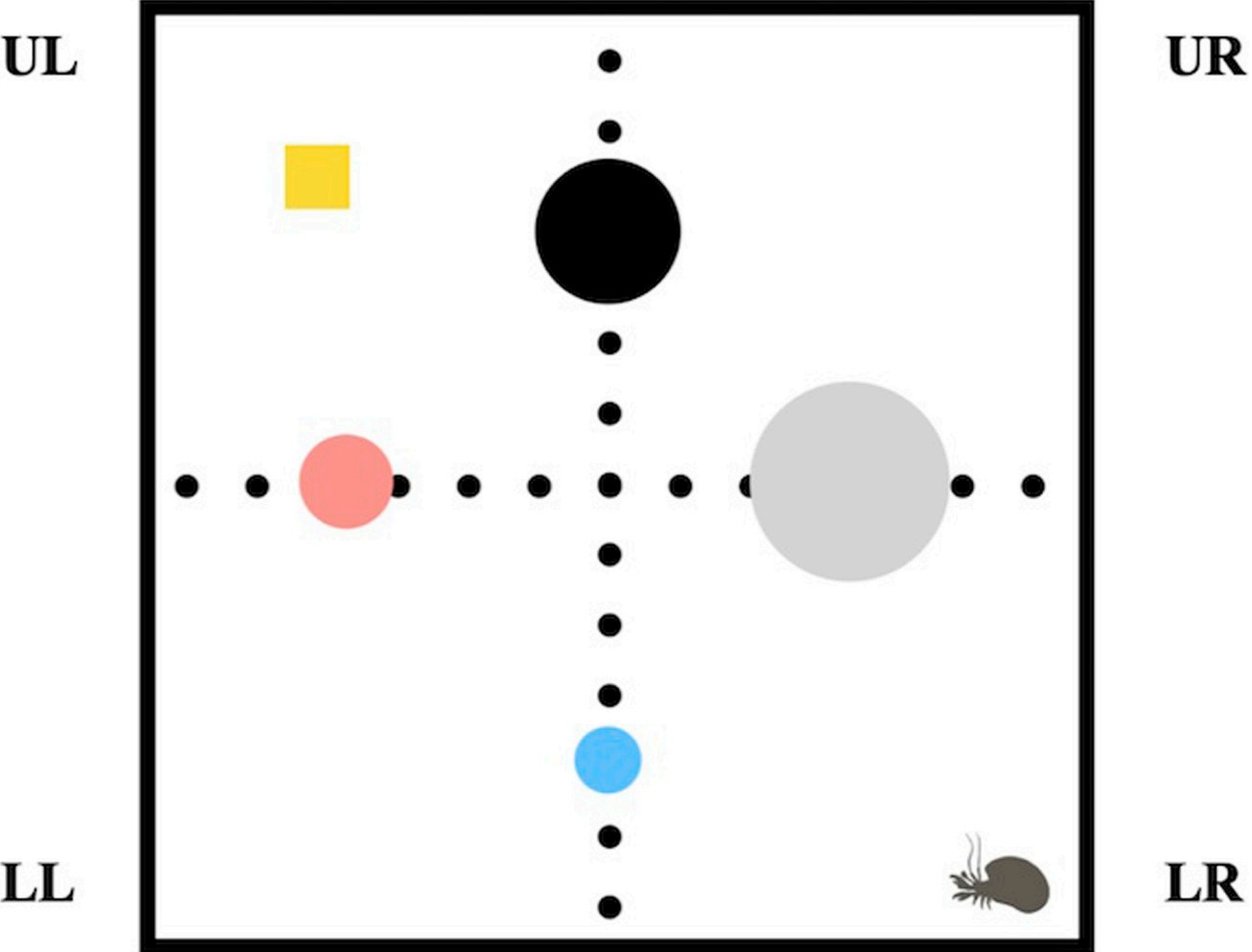

**Fig 1. Scheme of the experimental arena.** Diagram shows a bird's eye view of the arena, quadrants' limits (dotted lines) and visual cues' positions (colored circles). Crab silhouettes indicate the alternating starting points used throughout training trials and the yellow square shows the target location (in the upper left (UL) quadrant). Other quadrants named as upper right (UR), bottom left (BL), and bottom right (BR).

the starting point was changed in each trial (excluding the UL quadrant where the food was placed, see Fig 1) in a semi-random sequence as follows: UR, BL, BR, BL, UR, BR, UR, BL. In addition, the tray with food was placed below the sand surface, so that it would only be visible when the animals were very close (1 cm), thus the crabs were able to locate it from afar using only olfactory stimuli and the visual cues from the arena. The short-term memory retrieval test was carried out in the same way as the training trials, except that the food was withdrawn and the starting point was in the BR quadrant. This test was performed 5 minutes after the last training trial. Finally, the same experimental protocol was followed for long term-memory retrieval testing on Days 2, 4 and 7 after training (i.e., 24, 72 and 168 h after the last conditioning trial) (Fig 2).

### 2.4 Statistical analysis

The effect of training on the time to solve the task was evaluated using a generalized linear mixed-effect model or GLMM [27], where the time need to reach the tray was modeled as a

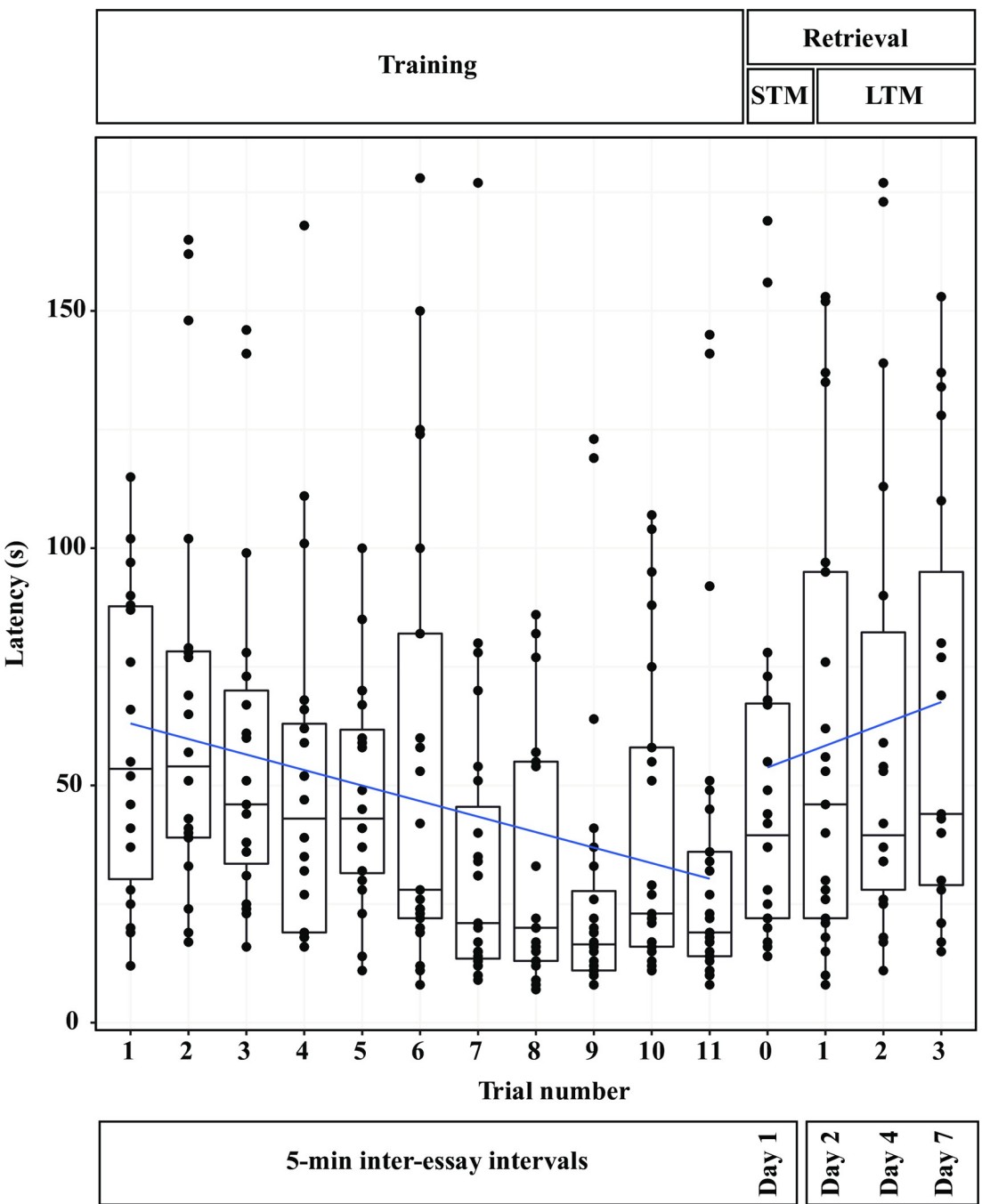

**Fig 2. Latency to reach the reward location during learning and testing.** Solid blue lines represent a function describing the training and retrieval phases of *n* = 26 hermit crabs as modeled by a generalized linear mixed-effect model (GLMM). Points (closed circles) represent latency for each crab. Median and interquartile ranges latency in seconds are presented. Decrease in latency to reach the target during training and increase during short- and long-term memory retrieval were significant.

Poisson process. GLMMs allows to specify a family parameter that better matched the distribution of the response variable, which in this case was composed of positive integers with a heavy left skew. Models were ran using time evaluated as categorical (trial number) or continuous variable (hours elapsed between trials) as the main effect. Data were grouped by individual

crab, fitted as random intercepts. The significance of fixed effects was tested using likelihood ratios tests. We inspected diagnostic plots for the model to find that while residuals were symmetrically distributed, they were heteroscedastic with a bias towards lower values. In addition, time spent exploring in the quadrant that originally housed the reward vs all others were compared by post hoc Tukey tests. We provide medians and interquartile ranges (IQR) or mean and standard error of the mean (SEM) in text and figures. Statistical analyses were performed using R [28] and GraphPad Prism 9 [29]. Significance was set at $p < 0.05$.

## Results

All crabs tested were able to locate the food in at least 9 of the 15 trials. Individuals that did not reach the tray within 180 sec were not included in the statistical analysis in that particular trial. Over the course of 11 trials, crabs were trained to locate the food reward. We observed a negative and significant decrement in the time to solve the task during this acquisition phase (Est = $-0.09 \pm 0.02$, $p < 0.001$), indicating that the crabs learned the location of the food and reached the hidden tray quicker as the number of trials increased (Fig 2). After the food was removed in the short-term, as well as in the three long-term memory tests we observed a positive and significant increase in the time needed to reach the tray (Est = $0.11 \pm 0.05$, $p = 0.03$), suggesting that crabs were using olfactory cues to find the food reward and were not able to reach its placement quickly in the absence of food (Fig 2). A significant increase of the time to find the food location was found between the last training trial and the short-term memory test (median = 19, IQR = 14–36 vs median = 39.5, IQR = 22–67.25, W = 26, $p = 0.003$). In the same way as for latency, the average distance traveled to reach the reward decreased during training and subsequently increased in the memory retention tests. The slope between the distance travelled and the training trials was negative but not significant (Est = $-0.08 \pm 0.02$, $p > 0.05$). Similarly, the slope between the distance travelled and the retrieval trials was positive but not significant (Est = $0.1 \pm 0.06$, $p > 0.05$) (see S1 Fig). As for in latency analysis, we excluded the trials where the crabs did not reach the food tray. However, in the single short-term and the three long-term memory retrieval testing, crabs spent significantly more time exploring in the UL quadrant than in any other quadrant of the arena. Three out of four ANOVAs comparing percent of activity between quadrants were significant (Day 1: $F(3) = 18.06$, $p < 0.001$; Day 2: $F(3) = 6.81$, $p < 0.001$, Day 4: $F(3) = 2.22$, $p = 0.09$; Day7: $F(3) = 13.61$, $p < 0.001$); post hoc Tukey tests, revealed that the comparison between the UL and all other quadrants were significant for Day 1 and for Day 7; only BR vs UL and UR vs UL for Day 2, and UR vs UL for Day 4 (Fig 3). Additionally, when in the UL quadrant, they spent most of the time above the precise location where the food reward was hidden (same location as during training) (Day 1: $62.70 \pm 18.49\%$; Day 2: $67.95 \pm 16.85\%$; Day 4: $76.01 \pm 14.71\%$; Day 7: $70.3 \pm 19.94\%$). The percentage of time spent in the tray's former place spot indicates that even in the absence of food reward, *C. compressus* could find its location, presumably based only on the visual cues of the experimental arena they had retained.

## Discussion

In the current study we evaluated the behavioral response of terrestrial hermit crabs in an allocentric spatial learning task using novel food as reinforcement. Thus, we assessed the strategy used by these animals to solve the task and their learning capabilities in a single multi-trial acquisition session; we also determined whether spatial memory persists after one sort-term and three long-term memory tests without the food reinforcement. Crabs were successfully trained to search and locate the food reward, as evidenced by the significant decrease in the time they required to find it until reaching an asymptotic performance. This was probably due

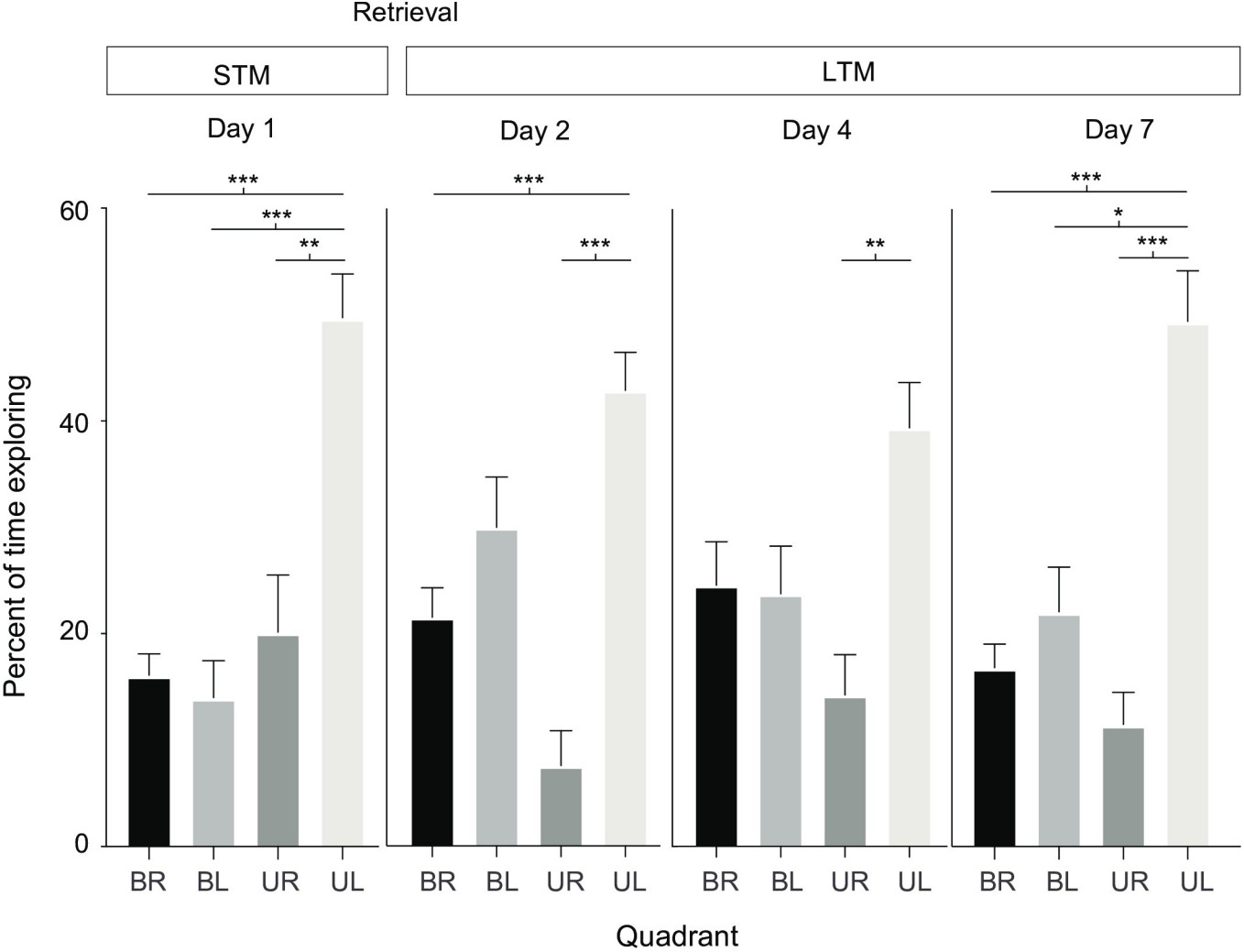

**Fig 3. Percent of time exploring per quadrant.** Mean percentage ± SEM of time spent actively exploring each quadrant for tests done on Days 1, 2, 4 and 7. Significance is specified as *p < 0.05 and **p < 0.01, **p < 0.001, as per the results of a Tukey test.

to the use of a highly palatable stimulus (peanut butter was the crabs' most preferred food in the pilot study), which is consistent with Klappenbach et al. [24] and Elwood & Appel [30], in the sense that animals' motivational state is crucial for behavioral expression during training and memory retrieval. Thus, most of the experimental protocols using appetitive food reinforcement are usually associated with chronic food deprivation [31]. In the present study however, as peanut butter acted as a novel and very rewarding stimulus, a short three-day fasting period prior the single training session was enough to elicit persistent seeking behavior for eleven consecutive trials apparently driven by a sustained motivation to find and consume the food reinforcer. This finding is in line with Tran [32] who reports that novel food items represent valuable resources for generalist scavengers, such as hermit crabs, because of the nutritional supplements they can provide, making them an adaptive foraging strategy. Moreover, it is worth noting that chemosensory detection of the reward (peanut butter odor) played an important role in the crabs' drive to search during the training phase and it was undoubtedly used to guide them to the target. This was evident in the retrieval testing trials without food, since from the short-term memory test performed only 5 min after the last training trial, the

latency to reach the place where the food was located significantly increased (as compared by a Wilcoxon test: W = 26, p = 0.003). This finding could be interpreted as an indication that crabs were not able to learn the spatial food location, however the fact that they still reached the precise place where the food was located in a shorter time than at the beginning of the training phase together with the relevant finding that crabs remained a greater percentage of time in the target quadrant (UL) and particularly in the exact food place spot, clearly indicates that animals remembered the reward location and could reach the target using solely the spatial cues found in the arena. Our data showed that when exposed repeatedly to the same task crabs were able to find the food in progressively shorter times which matches with prior observations, since early reports, e.g., Yerkes [33], who found that Brachyuran *Carcinus maenas* learned to avoid blind alleys on a simple labyrinth path, to recent studies that have confirmed and extended them [23, 34, 35]. Remarkably, eighty percent (18 of 26) of the crabs solved the task since the first trial and all of them showed consistent downward trends in the time employed to locate the food. However, this consistency was not observed in the distance traveled to reach the food, despite the fact that it also decreased with the progress of the training trials, indicating that the crabs improved their search strategy without optimizing it using only an allocentric spatial orientation. This is in line with our findings from short- and long-term memory tests showing that animals could arrive at the exact location of the reward, but required more time and traveled more distance than during the last training trials.

Unlike previous work, in the current study we decided to train the crabs in a single acquisition session in order to build up the learning curve based exclusively on their working memory and to assess short- and long-term memory separately. This allowed us to realize that although the olfactory component plays an important role in tracking the peanut butter to locate it, the visual clues of the arena were enough to reach the exact place where the reward was in its absence. Likewise, we were able to verify that although the time taken to reach the reward site gradually increased in the three long-term retrieval tests, the search for and permanence of the crabs in the target quadrant did not decrease, indicating a preserved spatial memory.

On the other hand, the current results reveal that after a single training session of eleven trials crabs consolidate a persistent long-term spatial memory, revisiting the spot where the food was hidden even seven days after it was removed. Brachyuran crustaceans [23] and hermit crabs [32] also retain for several days their attraction to known odors that have accurately predicted food availability in the past. Therefore, it is not surprising that *C. compressus* showed a robust long-term memory as consequence of the repeated reinforcing events experienced during conditioning. Studies in the crayfish *Procambarus clarkii* [36] and the crab *Chasmagnathus granulatus* [37] have evidenced the existence of both short- and long-term memory systems in crustaceans and some of its neurological basis, while other studies suggested long-term social memory capabilities in Anomurans [38]. However, our findings regarding extinction were unexpected, since we had hypothesized that crabs would quickly extinguish searching behavior in the absence of reward as they no longer obtained the reinforcer in the place where it was found, acquiring this new information and updating their memory. Interestingly, while the increased latency could be interpreted as an extinction process, the persistence in seeking the reward where it was previously found indicates the opposite and, in fact, suggests that the crabs do not extinguish the original learning, showing little behavioral flexibility, as occurs with social memory [36, 38]. This finding contrasts with previous studies showing clear evidence in crabs that long-term associative memories exhibit both extinction and reconsolidation processes depending on the duration of the reminder (i.e., re-exposure to the conditioned stimulus) [39–41].

It is important to highlight that the main difference between the training session and the short- and long-term memory tests was the absence of the reinforcement which, as mentioned

above, seems to have affected crabs' behavior for reasons other than learning itself, causing the increase in latency. It is difficult to estimate the contribution of the lack of reinforcement, on the one hand, and an extinction process on the other, in the gradual latency increase to reach the exact place where the food was located, since the most reliable parameter to assess memory retention (i.e., time spent exploring in the target quadrant) changed little on all four memory tests (one short-term and three long-term), showing a gradual decline on Days 2 and 4 but recovery on Day 7, suggesting that crabs had a well-preserved spatial memory.

Like other taxa, hermit crabs possess innate sensory mechanisms that allow them to discriminate between edible and inedible novel items upon the first encounter without the requirement of associative learning [32, 33]. In the context of foraging, novel food reinforcement takes place when, after sensing it, the animal consumes it [32]. Therefore, it was important to let the crabs to eat the peanut butter during the conditioning session to maintain the baseline level of motivation and consequently responding to the food stimulus. Coenobitids (Anomura) have evolved a good aerial sense of distance olfaction that is anatomically centered on the first antenna and behaviorally highly relevant [42], detecting food as far as 5 m [43, 44], however there is some controversy about the existence of an odoriferous chemical substances release system that could be left as a trace in the substrate for spatial orientation in this group [22]. Though, when in threatening situations, hermit crabs use multisensory channels to evaluate their environment [45], which has also been observed in other crustaceans as fiddler crabs (*Uca vomeris*) [46]. More studies are needed to determine the sensory inputs used in this group for the resolution of spatial tasks.

Coenobitids are attracted preferentially by the odors of foods that they have not eaten recently rather than foods eaten in the previous feeding event [47], even when these events take place in the last 24 hours [48]. This was not observed in our study, perhaps because animals have been deprived from food three days before the beginning of experiment so they could be more eager to eat and also, or because they did not have any other food option than peanut butter. In some insects, such as locusts, it has been noticed that stimulus strength depends on the internal state of animals, such as hunger levels [49].

One novelty of our visuospatial design that contrasts with other allocentric spatial learning tasks was that the visual cues were immersed on the experimental arena as it occurs with landmarks in the wild and manipulates variables that are relevant to crabs' modes of adjustment. This intended to be an analog paradigm to hermit crabs natural foraging routines, in which the accessibility of resources could be cyclical but some landmarks are maintained for long periods of time. *C compressus* may performed well because of the learning processes involved might be used in tasks encountered by crabs naturally and reflect essential features of the behavior occurring in the natural environment [31, 50, 51].

Despite the large body of studies evaluating learning abilities of Anomuran crustaceans, e.g., shell exchange [52–54], conspecific recognition [38, 55], social information transmission [56], food preference [48], solution of novel situations [57], and homing [58], there is no evidence of allocentric spatial learning in this crustacean order, even though the potential use of landmarks for spatial orientation (allocentric memory) has been previously suggested [23]. Few studies have shown plasticity in *C. compressus* learning simple "spatial" tasks, such as avoiding noxious stimuli [30] or decision making regarding external shell architecture on novel tasks [57]. However, none of them evaluate long-term memory. All these studies point out to *C. compressus* possessing an outstanding learning ability but were all performed over shorter time periods than in the present study, noticing the importance to explore complex relevant tasks such as spatial learning. Altogether our results suggest that the stimuli related with the reinforcement as well as the ones related with the environment influence the expression of behavior. This first description of allocentric spatial long-term memory in a terrestrial hermit

crab suggests a visuospatial memory formation. In nature, it is crucial for animals to learn and remember which places represent danger or which ones are associated with appetitive rewards such as food, shelter or mate [24]. A better appreciation of this adaptive trait in these animals will develop our understanding of resource exploitation by terrestrial crustaceans and their ecological roles as well as leading to potential comparative studies.

## Conclusion

We show that *C. compressus* was capable of solving a complex visuospatial learning task in addition to the other mentioned advantages which are demonstrated in malacostracan crustaceans as a potential model system. This first approach to spatial learning in a terrestrial hermit crab could lead to additional studies that let us have a better appreciation of this adaptive trait in these animals, such as individual variation factors (e.g., age, sex or personality) to predict whether and how the individuals locate the food in the most efficient way.

In the present study we show that the terrestrial hermit crab *C. compressus* is capable of learning the location of a palatable food stimulus using olfactory and visual cues, which was reflected in a gradual improvement in the execution of the task. Although this was an important finding because it implies that the standardization of the experimental conditions to evaluate the spatial abilities in this species were adequate, i.e., the motivation to execute and solve the test in a single multi-trial session, the most relevant finding was the fact that crabs retained a long-term spatial memory up to seven days based solely on visual cues, strongly suggesting an allocentric targeting strategy.

## Supporting information

**S1 Fig. Distance travelled from starting point to the reward location during learning and testing.** Solid blue lines represent a function describing the training and retrieval phases of *n* = 26 hermit crabs as modeled by a generalized linear mixed-effect model (GLMM). Points (closed circles) represent distance for each crab. Median and interquartile ranges distance in cm are presented. Decrease in distance to reach the target during training and increase during short- and long-term memory retrieval followed the expected trend but was non-significant. (TIF)

## Acknowledgments

This paper serves as fulfillment of ILVV for obtaining a PhD degree in Posgrado en Ciencias Biológicas, UNAM. We thank the Consejo Nacional de Ciencia y Tecnología de Mexico (CONACyT) for the graduate scholarship awarded to ILVV (630753). Finally, we appreciate the valuable comments of the reviews.

**Ethics in publishing**

Experiments were carried out in accordance with the National Institutes of Health guide for the care and use of laboratory animals.

## Author Contributions

**Conceptualization:** Ilse Lorena Vargas-Vargas, Jorge Contreras-Galindo, Gabriel Roldán-Roldán.

**Data curation:** Marcos Francisco Rosetti.

**Formal analysis:** Ilse Lorena Vargas-Vargas, Marcos Francisco Rosetti, Gabriel Roldán-Roldán.

**Funding acquisition:** Gabriel Roldán-Roldán.

**Investigation:** Ilse Lorena Vargas-Vargas, Gabriel Roldán-Roldán.

**Methodology:** Ilse Lorena Vargas-Vargas, Estefany Pérez-Hernández, Daniel González.

**Project administration:** Gabriel Roldán-Roldán.

**Resources:** Gabriel Roldán-Roldán.

**Software:** Marcos Francisco Rosetti.

**Supervision:** Marcos Francisco Rosetti, Gabriel Roldán-Roldán.

**Visualization:** Gabriel Roldán-Roldán.

**Writing – original draft:** Ilse Lorena Vargas-Vargas, Marcos Francisco Rosetti, Gabriel Roldán-Roldán.

**Writing – review & editing:** Ilse Lorena Vargas-Vargas, Daniel González, Marcos Francisco Rosetti, Jorge Contreras-Galindo, Gabriel Roldán-Roldán.

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
