## [Decision Letter · Decision Letter 0]

15 May 2023

PONE-D-23-07594EVIDENCE OF LONG-TERM ALLOCENTRIC SPATIAL MEMORY IN THE TERRESTRIAL HERMIT CRAB Coenobita compressusPLOS ONE

Dear Dr. Roldán-Roldán,

Thank you for submitting your manuscript to PLOS ONE. After careful consideration, we feel that it has merit but does not fully meet PLOS ONE’s publication criteria as it currently stands. Therefore, we invite you to submit a revised version of the manuscript that addresses the points raised during the review process.

We look forward to receiving your revised manuscript.

Kind regards,

Khor Waiho

Academic Editor

PLOS ONE

Journal Requirements:

"This paper serves as fulfillment of ILVV for obtaining a PhD degree in Posgrado en Ciencias Biológicas UNAM. We thank the Consejo National de Ciencia y Tecnología of Mexico (CONACyT) for funding and for the support of the research through a graduate scholarship to ILVV. "

"No. We specify the funding details in the Financial Disclosure section and include the sentence at the end of this section as requested. "

"Authors have NO competing interests."

Additional Editor Comments:

In light of the reviewers comments, I agree that the current manuscript needs major revision. In particular, the design of the behavioural experiment needs detailed explanation. For example, what measures were conducted to ensure that other factors such as visual or olfactory cues were being controlled.

Reviewers' comments:

Reviewer's Responses to Questions

**Comments to the Author**

1. Is the manuscript technically sound, and do the data support the conclusions?

Reviewer #1: Partly

Reviewer #2: Partly

2. Has the statistical analysis been performed appropriately and rigorously? 

Reviewer #1: Yes

Reviewer #2: No

3. Have the authors made all data underlying the findings in their manuscript fully available?

Reviewer #1: Yes

Reviewer #2: No

4. Is the manuscript presented in an intelligible fashion and written in standard English?

Reviewer #1: Yes

Reviewer #2: Yes

5. Review Comments to the Author

Reviewer #1: -- In the manuscript titled "EVIDENCE OF LONG-TERM ALLOCENTRIC SPATIAL MEMORY IN THE TERRESTRIAL HERMIT CRAB Coenobita compressus," the authors aim to shed light on the cognitive abilities of crustaceans, specifically with regards to their capacity for learning and spatial memory. The authors note that while some evidence of path integration and egocentric spatial learning has been observed in other crustaceans such as fiddler crabs and crayfish, there has been little research focused on the evaluation of allocentric learning in Anomuran crustaceans. Nevertheless, previous research suggests that this type of learning might exist in these animals. To investigate this, the authors designed an allocentric spatial task using a novel food reward for the Pacific hermit crab Coenobita compressus. The researchers evaluated the behavioral response of the hermit crabs in this task and measured their potential for learning in a single multi-trial acquisition session. Additionally, they assessed whether spatial memory persisted after short-term (30 minutes) and long-term memory tests (24, 48, and 72 hours) without the reward reinforcement. The authors report that the results of their study demonstrate that hermit crabs were able to learn the location of the reward through a combination of visuo-spatial and olfactory cue guidance. Specifically, C. compressus successfully located the food reward and retained spatial memory for up to 7 days after training. The authors highlight that their study represents the first description of allocentric spatial long-term memory in a terrestrial hermit crab. Certainly, this finding might provide new insight into the cognitive abilities of crustaceans and potentially have implications for further research in this field. The study design is straightforward and the methods, analysis, and presentation of data are sound, but there are a few issues that should be addressed. Overall, the manuscript is well written, but there are some areas where improvement is needed. I have provided specific suggestions below, and there may be additional areas where further refinement is necessary.

Major points:

- Why didn't the authors use untrained animals as a control? The utilization of this type of analysis, in place of training versus testing, allows for differentiation between the period of information acquisition and the period of assessment, as outlined by Rescorla in 1988, it is important to justify the authors` approach in both the Materials and Methods and Discussion sections. The authors should explain in the results contexts that animal behavior may differ between the training session and the three long-term memory (LTM) sessions for reasons unrelated to learning, which could either accentuate or obscure differences typical of memory processes.

-Authors selected two different parameters to evaluate Inter-trial training session (with the odor stimulus included) and the LTM behavioral output. It would be beneficial for the authors to provide a clear justification for this decision. It would also be useful to include analyses of both variables during the training and testing sessions. In this sense, the inclusion of both the time taken to reach the reward area and the percentage of total time spent exploring UL variables in the datasheet for both the TR and TS sessions would be highly beneficial.

-Due to the fact that the authors are presenting a new paradigm, it would be beneficial to include additional variables in their analysis. For example, they could consider including measures of distance such as the total walking distance traveled during the task.

-The potential influence of extinction or reconsolidation processes on the changes in behavioral outputs observed in the STM and LTM tests, especially with regard to the results shown in Figure 3, should be thoroughly addressed in both the 2.2 section and, more explicitly, in the discussion.

Minors

In the text, take into consideration that Davies et al 2019 denote that “The potential for allocentric (the use of landmarks) learning cannot be entirely discarded..”

The title of the variable of Figure 2 should be a description and not a conclusion.

Reviewer #2: The study by Vargas-Vargas et al. have studied allocentric learning in hermit crabs by evaluating the spatial learning abilities of Coenobita compressus. They first trained the crabs in a multi trial session using peanut butter as a palatable food source and placing 4 visual landmarks within an arena. They measure the time the crabs spend searching for food as well as the time they spend in specific area of arena during this multi trial session and then remove the food and do a short- and long-term memory test. As the crabs, search time decreased during muti-trial and then increased during long term memory trial they concluded that crabs are capable of learning the location of a palatable food stimulus using olfactory and visual cues. Although it could have been an interesting study, I am not convinced by their results mainly due to methodological flaws.

1- They did not control for visual cues. What I mean is that they never ran trials without visual cues or changing the location or order of visual cues to see if that affects crabs search for food. So, crabs search for food could be exclusively due to olfactory cues. It could be the trace of smell remains in the arena and as time passes the trace of smell reduces which can explain the increase in search time during long-term memory test. You also could have changed the location of your visual landmarks to see if that would change the location on the arena that the crabs search for food. It could be some other parameters that attracts crab to the UL quadrant.

2- The increase in search time during long-term memory test could be a new learning rather than not remembering. There isn't any food there anymore, so the crabs could learn there isn’t any food there so they take their time to search other area as well. You could have had three groups tested at different long-term intervals rather than all crabs in all intervals.

3- The starting point during training multi-trial does not seem to be balance across trials, there is no mention of it and in the first trial of this training all crabs started at LR quadrants. So the decrease in the search time could be due to different distances from food source.

4- The data they provided in figshare is not understandable, especially that column and row titles are not in English and It's not obvious which experiment they are representing.

5- Why did you use generalized mixed models rather than linear mixed effect models?

6. PLOS authors have the option to publish the peer review history of their article (what does this mean?). If published, this will include your full peer review and any attached files.

Reviewer #1: **Yes: **Alejandro Delorenzi

Reviewer #2: No

---

## [Author Response · Author response to Decision Letter 0]

1 Sep 2023

Reviewer Comments:

Reviewer #1: In the manuscript titled "EVIDENCE OF LONG-TERM ALLOCENTRIC SPATIAL MEMORY IN THE TERRESTRIAL HERMIT CRAB Coenobita compressus," the authors aim to shed light on the cognitive abilities of crustaceans, specifically with regards to their capacity for learning and spatial memory. The authors note that while some evidence of path integration and egocentric spatial learning has been observed in other crustaceans such as fiddler crabs and crayfish, there has been little research focused on the evaluation of allocentric learning in Anomuran crustaceans. Nevertheless, previous research suggests that this type of learning might exist in these animals. To investigate this, the authors designed an allocentric spatial task using a novel food reward for the Pacific hermit crab Coenobita compressus. The researchers evaluated the behavioral response of the hermit crabs in this task and measured their potential for learning in a single multi-trial acquisition session. Additionally, they assessed whether spatial memory persisted after short-term (30 minutes) and long-term memory tests (24, 48, and 72 hours) without the reward reinforcement. The authors report that the results of their study demonstrate that hermit crabs were able to learn the location of the reward through a combination of visuo-spatial and olfactory cue guidance. Specifically, C. compressus successfully located the food reward and retained spatial memory for up to 7 days after training. The authors highlight that their study represents the first description of allocentric spatial long-term memory in a terrestrial hermit crab. Certainly, this finding might provide new insight into the cognitive abilities of crustaceans and potentially have implications for further research in this field. The study design is straightforward and the methods, analysis, and presentation of data are sound, but there are a few issues that should be addressed. Overall, the manuscript is well written, but there are some areas where improvement is needed. I have provided specific suggestions below, and there may be additional areas where further refinement is necessary.

Major points:

1) Why didn't the authors use untrained animals as a control? The utilization of this type of analysis, in place of training versus testing, allows for differentiation between the period of information acquisition and the period of assessment, as outlined by Rescorla in 1988, it is important to justify the authors` approach in both the Materials and Methods and Discussion sections. 

Response: We understand the reviewer's concern for including an untrained control group that could allow us to differentiate behavioral changes during learning and during memory retrieval. However, in allocentric spatial memory laboratory models, control groups of untrained animals are not usually included because they do not perform the task, since there is no goal to reach or reward to obtain. In the case of aversive models that force the animal to escape (e.g., Morris Water Maze or Barnes Maze), control subjects are included, only to compare the effect of exercise or stress (1, 2). In appetitive learning models such as the Radial Arm Maze or the Buried Food Location Test (3,4), untrained controls are not used since their exploratory behavior is random and erratic or, as happens in fact, diminishes as the trials run, precisely because there is nothing to look for. In our experiment it was impossible to include untrained controls because unreinforced crabs simply did not investigate the testing arena. In fact, we carried out a series of pilot experiments with another batch of crabs in which we gently removed the animal from its shell and used its own shell as the reinforcer, but we were unsuccessful, the crabs did not show exploratory behavior probably because they were naked and vulnerable. We also tried other less palatable food reinforcers but they were not adequate to trigger searching behavior either, and the crabs remained where we left them at the beginning of the test or showed very little exploratory activity. 

On the other hand, it is worth noting that we did not make a comparison between training and testing strictly speaking, but rather analyze latency to reach the reward throughout the training phase, and then the time exploringpermanence in the reward quadrant of the arena duringin the short- and long-term memory tests. The only comparison we made between training and testing was the latency, which provided us key information about the role of the olfactory cue in finding the reward, but little information about memory itself (as discussed in the manuscript). Thus, what we are presenting in the Results section is the behavioral modification of the same group of crabs throughout different experimental conditions, which is the standard experimental design to analyze allocentric spatial learning. Basically, it consists of training the animals repeatedly to find a reinforcer until a learning criterion is reached. Subsequently, long-term memory is analyzed by removing the reinforcer and quantifying the time that the subject spends looking for the reinforcer in the place where it was found. Following the reviewer’s suggestion, we have added a clarifying paragraph about our experimental design in the Methods section.

2) The authors should explain in the results contexts that animal behavior may differ between the training session and the three long-term memory (LTM) sessions for reasons unrelated to learning, which could either accentuate or obscure differences typical of memory processes.

Response: The reviewer is right. Despite the fact that in our manuscript we mention that the odor of the reinforcer (peanut butter) was the main stimulus that clearly affected search behavior of crabs as evidenced by increased latency during short- and long-term memory testing, it is necessary to highlight that this is the most important difference between the learning session and the memory tests. Taking the above into account, we have referred to this fact in the Discussion section of the corrected manuscript, emphasizing the importance of the olfactory stimulus in the behavioral response of crabs. 

3) Authors selected two different parameters to evaluate Inter-trial training session (with the odor stimulus included) and the LTM behavioral output. It would be beneficial for the authors to provide a clear justification for this decision. It would also be useful to include analyses of both variables during the training and testing sessions. In this sense, the inclusion of both the time taken to reach the reward area and the percentage of total time spent exploring UL variables in the datasheet for both the TR and TS sessions would be highly beneficial.

Response: As could be inferred from the description of training and memory testing in point 2.2 in of the Methods section, it is impossible to measure the time spent in the target quadrant during the training phase because once the crabs reach the food, they stop exploring and start eating; in this point, the training trial ends and the crab is removed from the testing arena. On the other hand, the latency to reach the location of the reward, although it is a measurable parameter during memory retrieval testing (and in fact we report it in our results), it actually gives us less information than the time spentthe permanence in the target quadrant, since this last parameter indicates unequivocally that subjects remember where the reward was as they search insistently for it, while the latency can give us false positives, since crabs can pass through the site in question without really stopping to look for the reward. Consequently, it is essential to withdraw the reward stimulus to be able to assess both short- and long-term spatial memory, because it is the only way to isolate crab’s orientation based exclusively on visuospatial stimuli. However, the reviewer's questioning makes us think that this point was not sufficiently clear and therefore, following his suggestion, we have added explanatory paragraph in the Methods section.

-Due to the fact that the authors are presenting a new paradigm, it would be beneficial to include additional variables in their analysis. For example, they could consider including measures of distance such as the total walking distance traveled during the task.

Response: Following the reviewer's suggestion we have included the distance traveled to reach the reward in the supplementary figure 1. The slope between the distance travelled and the training trials was negative but not significant (Est = -0.08 ± 0.02, p > 0.05). Similarly, the slope between the distance travelled and the retrieval trials was positive but not significant (Est = 0.1 ± 0.06, p > 0.05). As for in latency analysis, we excluded the trials where the crabs did not reach the food tray.

-The potential influence of extinction or reconsolidation processes on the changes in behavioral outputs observed in the STM and LTM tests, especially with regard to the results shown in Figure 3, should be thoroughly addressed in both the 2.2 section and, more explicitly, in the discussion.

Response: The reviewer rightly brings up this important question and we have therefore addressed it in the Discussion section of the revised manuscript. The possibility that an extinction and/or reconsolidation process may have affected crabs’ behavior in the LTM trials iswas very high, since animals no longer obtained the reinforcer in the place where it was found and evidently, they were acquiring this new information and updating their memory. That is why we found it surprising that the crabs still reached the target site and remained there looking for food. We attributed this persistent behavior to increased hunger as crabs continued to fast.

Minors

In the text, take into consideration that Davies et al 2019 denote that “The potential for allocentric (the use of landmarks) learning cannot be entirely discarded.”

Response: At the reviewer's suggestion we have corrected that omission

The title of the variable of Figure 2 should be a description and not a conclusion.

Response: We have changed the title of the variable from “Solving time” to “Latency”

References: 

1. Barry D.N. & Commins S. A novel control condition for spatial learning in the Morris water maze. Journal of Neuroscience Methods 318 (2019) 1–5. https://doi.org/10.1016/j.jneumeth.2019.02.015

2. Gawel K, Gibula E, Marszalek-Grabska M, Filarowska J, Kotlinska JH. Assessment of spatial learning and memory in the Barnes maze task in rodents - methodological consideration. Naunyn-Schmiedeberg's Archives of Pharmacology (2019) 392:1–18. https://doi.org/10.1007/s00210-018-1589-y

3. Carrillo-Mora P, Giordano M, Santamaría A. Spatial memory: Theoretical basis and comparative review on experimental methods in rodents. Behavioural Brain Research 203 (2009) 151–164 doi: 10.1016/j.bbr.2009.05.022

4. Ramirez Ortega D, Ovalle Rodríguez P, Pineda P, González Esquivel D, Ramos Chávez LA, Vázquez Cervantes G, Roldán Roldán G, Pérez de la Cruz G, Díaz Ruiz A, Méndez Armenta M, Marcial Quino J, Gómez Manzo S, Ríos C, Pérez de la Cruz V. Kynurenine Pathway as a New Target of Cognitive Impairment Induced by Lead Toxicity During the Lactation. Scientific Reports (2020) 10:3184 https://doi.org/10.1038/s41598-020-60159-3 1

Reviewer #2: 

The study by Vargas-Vargas et al. have studied allocentric learning in hermit crabs by evaluating the spatial learning abilities of Coenobita compressus. They first trained the crabs in a multi trial session using peanut butter as a palatable food source and placing 4 visual landmarks within an arena. They measure the time the crabs spend searching for food as well as the time they spend in specific area of arena during this multi trial session and then remove the food and do a short- and long-term memory test. As the crabs, search time decreased during muti-trial and then increased during long term memory trial they concluded that crabs are capable of learning the location of a palatable food stimulus using olfactory and visual cues. Although it could have been an interesting study, I am not convinced by their results mainly due to methodological flaws.

1- They did not control for visual cues. What I mean is that they never ran trials without visual cues or changing the location or order of visual cues to see if that affects crabs search for food. So, crabs search for food could be exclusively due to olfactory cues. You also could have changed the location of your visual landmarks to see if that would change the location on the arena that the crabs search for food. It could be some other parameters that attracts crab to the UL quadrant.

Response: Certainly, the reviewer is correct that searching behavior could have been triggered and guided by the prominent olfactory stimulus of the reward (peanut butter), that we do not dispute and indeed discuss in our manuscript. However, the significant decrease in the latency to find the reward throughout the training session cannot be explained exclusively by olfactory-guided behavior. In fact, our experimental design is based on previous studies carried out to evaluate olfactory acuity in rodents, which show precisely that changing the place where the reward is located does not cause a decrease in the latency to find it, but rather keeps it constant (1). This is equivalent to moving the visual cues around the arena or removing them entirely, since with this manipulation the spatial reference is completely lost. On the contrary, keeping the reward and the visual cues in the same place and changing the starting point of the experimental subject in each trial, causes a reduction in latency attributable to spatial learning (2). In other words, when the subject uses only the olfactory gradient emanating from the stimulus as a guide to find it, it depends exclusively on its olfactory acuity, and this does not change throughout the trials. This is what happens when the olfactory stimulus is moved around in the test arena. On the contrary, when the stimulus and the landmarks remain constant and what is changed is the starting point in each trial, the subject locates the stimulus site in space and gradually forms a spatial memory that is reflected in the decrease of latency and the distance traveled to find it. The most widely used models to assess allocentric spatial learning are based on this principle, such as the Morris Water Maze or the Barnes Maze, as well as tests that use food as a reward, such as the Radial Arm Maze or the Buried Food Location Test (2,3,4). Furthermore, the fact that during the memory recall tests, both short and long term, the olfactory stimulus was absent, leads us to the conclusion that the only possibility to reach the precise place where the reward was and to search for it insistently, is through a spatial memory based on the visual keys in the arena.

-It could be the trace of smell remains in the arena and as time passes the trace of smell reduces which can explain the increase in search time during long-term memory test. 

Response: Throughout the experiment we try to eliminate, or at least minimize, the possibility that an olfactory trace left by the animal or the food would interfere with spatial learning. In the Methods section of the manuscript, it is explained that between each of the trials during the training session, the sand from the test arena was completely mixed up and rehydrated, in order to eliminate any olfactory trace left by the crab in the previous trial. On the other hand, the food was never in contact with the sand but rather in a small plastic tray, which also prevented it from leaving any olfactory remnant. During the short- and long-term memory tests the same procedure was followed but, in addition, the food was absent. Therefore, we believe that there is no reason to suppose that there was an olfactory trace that remained in the sand (neither from the crab nor from the food) that could have guided the crab throughout the learning process, much less during memory recall testing. In addition, the increase in the search time of the reinforcer during the long-term memory tests did not occur randomly but in the target quadrant (see Fig. 31 attached). In the last paragraph of point 2.3 of the Methods section, it is explained that the short- and long-term memory retention tests were done in the same way as during training, but without reward, that is, they all lasted 180 seconds in which the crab roamed freely in the testing arena, and what is presented in the Results is the percentage of time in each quadrant.

2- The increase in search time during long-term memory test could be a new learning rather than not remembering. There isn't any food there anymore, so the crabs could learn there isn’t any food there so they take their time to search other area as well. 

Response: The increase in search time occurred only in the reward quadrant, not in the other quadrants, as can be seen in Figure 3 of our manuscript, in which we now plot the percentage of time spent exploring in all quadrants, as opposed to the old version which showed opposing quadrants as conventionally reported in allocentric spatial memory studies. However, the point that the reviewer brings up is very interesting and indeed that was what we expected to happen, i.e., an extinction process based on new learning. Though, judging by the data, this was not the case. Certainly, the latency to get to the exact site where the reinforcer was augmented gradually, but the permanence in the target quadrant always remained high. This indicates, according to us, that crabs remembered where the reinforcer was and, unlike what happens in other species such as rodents, crabs do not extinguish easily nor do they seem to show behavioral flexibility, but instead persist in maintaining behavior based on the original memory. We have added a paragraph in the discussion to refer to this interesting aspect.

-You could have had three groups tested at different long-term intervals rather than all crabs in all intervals.

Response: Indeed, as the reviewer rightly points out, one of the ways to measure long-term memory retention over time is what he suggests, particularly when trying to eliminate the effect of extinction. The reasons for using the same animals in the 3 long-term memory tests of our study were two: 1) The availability of animals; in behavioral studies with invertebrates, relatively large sample sizes are used, so we prefer to evaluate 1 large group instead of 3 small groups. 2) In our case, what we wanted to evaluate in addition to retention was the extinction process and for this reason we used the same group. As stated in the last paragraph of the Introduction section, we hypothesized that the crabs would extinguish quickly but, as this was not the case, we extended the intervals between trials. To our surprise, the crabs did not extinguish what they had learned, at least not when measuring our most reliable parameter, which was the permanence and/or search in the place where the reward was found. We suspended the experiment because the crabs had not eaten for 10 days, a factor that also seems to have influenced their behavior, as discussed in the Manuscript.

3- The starting point during training multi-trial does not seem to be balance across trials, there is no mention of it and in the first trial of this training all crabs started at LR quadrants. So, the decrease in the search time could be due to different distances from food source.

Response: Certainly, in the Methods section, this aspect mentioned by the reviewer was not clear. This omission has been corrected (see point 2.3) and now it is specified how the starting point was balanced. On the other hand, there was no such decrease in target quadrant search time during long-term memory tests (see new Fig. 34). On the contrary, unexpectedly, the time spent by the crabs searching for the reward in the target (UL) quadrant remained high (see Fig 1 below).

4- The data they provided in figs. hare is not understandable, especially that column and row titles are not in English and It's not obvious which experiment they are representing. 

Response: The data in figshare was indeed unclear. We have translated all the text in the files into English. Also, we have renamed the variable names so that they match the names in the manuscript. We hope these modifications have made it more understandable to determine which column represents which variable of the experiment.

5- Why did you use generalized mixed models rather than linear mixed effect models? 

Response: The reviewer is correct. Since we assumed a normal distribution for our response variable, a linear mixed model is a more accurate description of the statistical method we used. We have amended the text accordingly (see point 2.4 of the Methods section).

References:

1. Lehmkuhl, A. M., Dirr, E. R. & Fleming, S. M. Olfactory assays for mouse models of neurodegenerative disease. J Vis Exp, e51804,

https://doi.org/10.3791/51804 (2014). 

2. Sharma S, Rakoczy S, Brown-Borg H. Assessment of spatial memory in mice. Life Sci. 2010 October 23; 87(17-18): 521–536. doi:10.1016/j.lfs.2010.09.004. 

3. Carrillo-Mora P, Giordano M, Santamaría A. Spatial memory: Theoretical basis and comparative review on experimental methods in rodents. Behavioural Brain Research 203 (2009) 151–164 doi:10.1016/j.bbr.2009.05.022 

4. Ramirez Ortega D, Ovalle Rodríguez P, Pineda P, González Esquivel D, Ramos Chávez LA, Vázquez Cervantes G, Roldán Roldán G, Pérez de la Cruz G, Díaz Ruiz A, Méndez Armenta M, Marcial Quino J, Gómez Manzo S, Ríos C, Pérez de la Cruz V. Kynurenine Pathway as a New Target of Cognitive Impairment Induced by Lead Toxicity During the Lactation. Scientific Reports (2020) 10:3184 https://doi.org/10.1038/s41598-020-60159-3 1

---

## [Decision Letter · Decision Letter 1]

25 Sep 2023

PONE-D-23-07594R1EVIDENCE OF LONG-TERM ALLOCENTRIC SPATIAL MEMORY IN THE TERRESTRIAL HERMIT CRAB Coenobita compressusPLOS ONE

Dear Dr. Roldán-Roldán,

Thank you for submitting your manuscript to PLOS ONE. After careful consideration, we feel that it has merit but does not fully meet PLOS ONE’s publication criteria as it currently stands. Therefore, we invite you to submit a revised version of the manuscript that addresses the points raised during the review process.

We look forward to receiving your revised manuscript.

Kind regards,

Khor Waiho

Academic Editor

PLOS ONE

Journal Requirements:

Reviewers' comments:

Reviewer's Responses to Questions

**Comments to the Author**

1. If the authors have adequately addressed your comments raised in a previous round of review and you feel that this manuscript is now acceptable for publication, you may indicate that here to bypass the “Comments to the Author” section, enter your conflict of interest statement in the “Confidential to Editor” section, and submit your "Accept" recommendation.

Reviewer #1: All comments have been addressed

2. Is the manuscript technically sound, and do the data support the conclusions?

Reviewer #1: Yes

3. Has the statistical analysis been performed appropriately and rigorously? 

Reviewer #1: Yes

4. Have the authors made all data underlying the findings in their manuscript fully available?

Reviewer #1: Yes

5. Is the manuscript presented in an intelligible fashion and written in standard English?

Reviewer #1: Yes

6. Review Comments to the Author

Reviewer #1: While the questions raised have been addressed, please consider including in the discussion that, in the case of crustaceans, there is indeed evidence that long-term associative memories exhibit both extinction and reconsolidation processes (Pedreira ME et al. 2004; Merlo E et al. 2008; Hepp et al. 2010). In light of this, it is advisable to discuss the highlighted paragraphs in R1 (lines 261-268 and 298-306) collectively rather than separately in the text.

7. PLOS authors have the option to publish the peer review history of their article (what does this mean?). If published, this will include your full peer review and any attached files.

Reviewer #1: No

---

## [Author Response · Author response to Decision Letter 1]

10 Oct 2023

We have added the suggested paragraph in the discussion, as can be seen in the version with tracked changes. This addition further improved the discussion and we appreciate the reviewers input on the matter.

---

## [Editor Report · Decision Letter 2]

11 Oct 2023

EVIDENCE OF LONG-TERM ALLOCENTRIC SPATIAL MEMORY IN THE TERRESTRIAL HERMIT CRAB Coenobita compressus

PONE-D-23-07594R2

Dear Dr. Roldán-Roldán,

We’re pleased to inform you that your manuscript has been judged scientifically suitable for publication and will be formally accepted for publication once it meets all outstanding technical requirements.

Kind regards,

Khor Waiho

Academic Editor

PLOS ONE

---

## [Editor Report · Acceptance letter]

16 Oct 2023

PONE-D-23-07594R2 

Evidence of long-term allocentric spatial memory in the Terrestrial Hermit Crab *Coenobita compressus*

Dear Dr. Roldán-Roldán:

I'm pleased to inform you that your manuscript has been deemed suitable for publication in PLOS ONE. Congratulations! Your manuscript is now with our production department. 

Kind regards, 

on behalf of

Dr. Khor Waiho 

Academic Editor

PLOS ONE